

# Factors influencing compliance with endoscopy final rinsing water standards: a study in a tertiary hospital setting

Yuhua Yuan[*], Lihong Ye[*], Tianyi Lu, Baihuan Feng and Jin Zhao

Department of Infection Prevention and Control, Sir Run Run Shaw Hospital, Zhejiang University School of Medicine, Hangzhou, China
[*] These authors contributed equally to this work.

## ABSTRACT

**Background**. Maintaining a high compliance rate for final rinsing water is essential for patient safety and infection control in healthcare facilities. This study aims to investigate the causes of fluctuations in compliance rates at a tertiary hospital and evaluate the effectiveness of preventive measures.

**Methods**. Monitoring data from October 2022 to December 2023 were analyzed to assess microbial contamination in final rinse water. Environmental assessments and literature reviews were conducted to identify potential contributing factors. Strategic interventions were implemented, and their impact on water quality and infection control was systematically evaluated.

**Findings**. Initial assessments revealed fluctuating compliance rates, with faucet aerators identified as a potential source of contamination. Water samples collected before the aerator showed 100% compliance (6/6 samples), while colonies of *Cupriavidus pauculus* and *Stenotrophomonas maltophilia* were detected on aerator surfaces and in final rinse water. After removing the aerators and enhancing disinfection protocols, compliance improved significantly, with subsequent samples meeting infection control standards ($\leq$10 CFU/100 mL).

**Conclusion**. This study suggests a potential link between fluctuations in endoscopy final rinsing water quality and the presence of faucet aerators. These findings support further research and the development of guidelines for the appropriate use of aerators in healthcare settings.

Corresponding authors
Baihuan Feng,
fengbaihuan@zju.edu.cn
Jin Zhao, enqizhao@zju.edu.cn

## BACKGROUND

Since the late 1970s, numerous studies have documented healthcare-associated infection (HAI) linked to endoscopic procedures (*Kovaleva et al., 2013*), with affected patients often experiencing a notably high mortality rate (*Swei et al., 2022*; *Verfaillie et al., 2015*). Adverse clinical outcomes, including infections caused by fungi, Legionella spp., environmental mycobacteria, *Pseudomonas* spp., and *Salmonella* spp., have been consistently attributed to biofouling in endoscope rinse water (*Bajolet et al., 2013*; *Goyal et al., 2022*). In turn, this contamination has led to pseudo-outbreaks, which present significant clinical diagnostic

and management challenges (*Murdani et al., 2017*). Collectively, these findings highlight the critical need to focus not only on the overall cleanliness of endoscopic equipment but also on the microbial quality of the rinse water used throughout the reprocessing workflow.

Understanding and optimizing endoscopic rinsing procedures is essential for effectively mitigating healthcare-associated infections associated with endoscopic interventions. This comprehensive processing workflow encompasses several critical stages: pre-processing, cleaning, disinfection, and final rinsing. While current international guidelines, such as those from Association for the Advancement of Medical Instrumentation (AAMI) (*Association for the Advancement of Medical Instrumentation, 2021*), recommend automated endoscope reprocessors (AERs) as the preferred method due to their standardized and reproducible performance, manual reprocessing remains widely practiced in many healthcare settings, particularly in resource-limited or high-workload environments. Importantly, the microbial and chemical quality of rinse water has been shown to critically influence the final cleanliness and disinfection efficacy of endoscopes (*Marek et al., 2014*; *Joint Working Group of the Hospital Infection Society (HIS) & Public Health Laboratory Service (PHLS), 2002*).

In the pre-processing, the primary goal is to remove gross organic and inorganic debris. This is followed by a meticulous cleaning phase, in which specialized brushes and enzymatic or detergent-based solutions are used to achieve complete physical decontamination. Next, high-level disinfection is performed to eliminate pathogenic microorganisms, representing a key step in preventing cross-contamination. The final rinsing step ensures the removal of any residual disinfectants or cleaning agents, thereby minimizing the risk of chemical irritation or toxicity during clinical use.

Emphasizing the water quality in the final rinsing stage is paramount, as it directly correlates with the endoscope's cleanliness and patient safety. Several international and national guidelines (*Ministry of Health of the People's Republic of China, 2022*; *International Organization for Standardization, 2024*; *National Health Family Planning Commission of the People's Republic of China, 2016*), including ISO 15883 (*Speer et al., 2023*), China's WS 507-2016 (Technical Specification for Cleaning and Disinfection of Flexible Endoscopes), have established rigorous microbial contamination thresholds for final rinse water (*National Health Family Planning Commission of the People's Republic of China, 2016*). This standards highlight the critical role of water quality in endoscope reprocessing, as maintaining compliance is not only a regulatory requirement but also a key measure for reducing the risk of healthcare-associated infections. Therefore, strict adherence to these guidelines is essential to ensure the safety and efficacy of endoscopic procedures throughout the entire clinical workflow.

In a recent investigation at Sir Run Run Shaw Hospital, Zhejiang University School of Medicine, marked variability was identified in the compliance rate of final rinse water microbiological quality. Based on a thorough analysis of longitudinal monitoring data, environmental surveys, and pertinent literature, this study systematically investigates the root causes underlying this inconsistency. Through the strategic implementation of targeted interventions and meticulous evaluations, it aims to effectively address identified contamination risks and propose evidence-informed strategies for strengthening water

quality control. By ensuring the sustained delivery of microbiologically safe rinse water, our findings emphasize the critical importance of data-driven practices in optimizing infection prevention and enhancing patient safety in endoscopic care. These results contribute novel insights to the ongoing discourse on endoscope reprocessing protocols and water safety standards in clinical settings.

## MATERIALS & METHODS

### Subjects

According to national standards, Sir Run Run Shaw Hospital, Zhejiang University School of Medicine conducted monthly testing of medical water. In cases where the monitoring results were not satisfactory, resampling was performed until compliance was achieved, including for the final rinsing water. The data for this study were derived from the microbiological monitoring results of final rinsing water collected by the Infection Control Team between October 2022 and September 2023. Since some locations were tested multiple times in certain months, only the results of the initial test for each location were included in this study to represent the baseline microbial status prior to any corrective actions. Including all data points could have confounded the evaluation of the original contamination source. In total, the study incorporated microbiological data from final rinsing water samples taken from six manually cleaned sinks.

### Hospital reconstruction

The Sir Run Run Shaw Hospital, Zhejiang University School of Medicine was expanded from August 2022, which included renovation of the water supply system. During this renovation aerator were connected to most faucets of the water supply in the hospital wards. The purified water was distributed to each floor from the hospital's water storage tank through the hospital's plumbing system. To ensure water quality, the entire system, including the storage tank, was maintained according to national guidelines. Additionally, weekly waterline disinfection was performed using health essence disinfecting effervescent tablets at a concentration of 500 mg/L, in strict accordance with the manufacturer's specifications.

### Environmental investigation

In September 2023, multiple re-sampling was conducted for all the artificial pipelines that failed the initial monitoring. Additionally, pipeline disinfection was carried out every Monday following each sampling. The initial sampling locations included water collected before aerator, the surface of aerator, and final rinsing water, with subsequent samplings focusing solely on the final rinsing water. Furthermore, the infection control team conducted on-site inspections of the pipeline, water supply center, and their working environment at hospital.

### Water quality monitoring and testing

The study included testing three sample types: final rinsing water, water collected before aerator, and the surface of aerator (Fig. 1). During sampling, we firstly collected 100
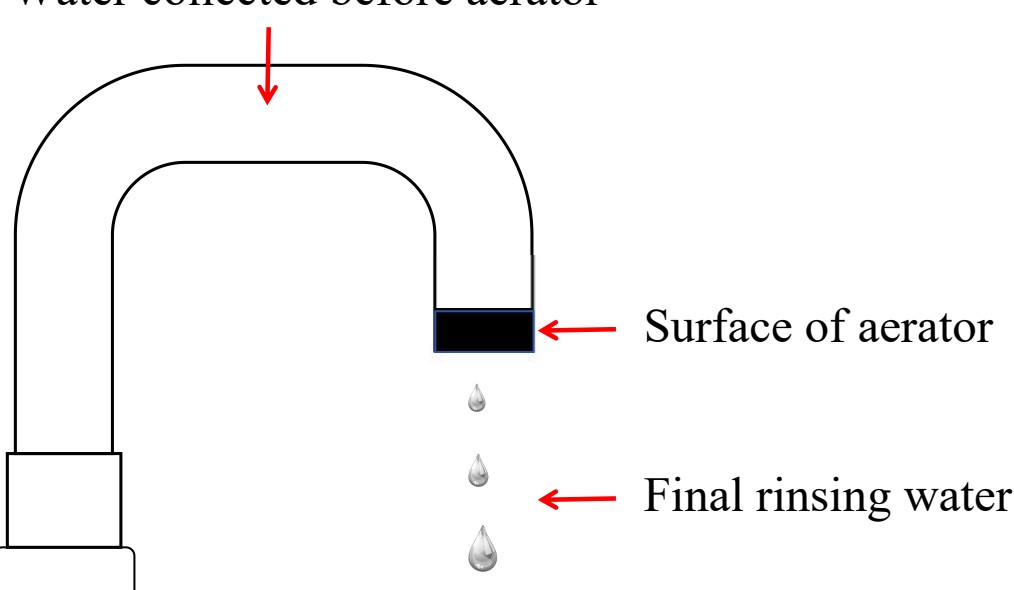

**Water collected before aerator**

**Surface of aerator**

**Final rinsing water**

**Figure 1  Schematic diagram of sampling locations for endoscope rinsing water.**

mL final rinsing water which underwent filtration using a membrane with a pore size of 0.45 μm followed by incubation at 36 ± 1 °C for 48 h. After then aerator was removed from the faucet, it was immersed in 100 mL of elution fluid with gentle shaking. Concurrently, we collected 20 mL water before aerator. one mL elution fluid or water before aerator was cultured in a Petri dish containing 15 mL of nutrient agar, and a corresponding blank control plate was meticulously prepared. These Petri dishes were subsequently incubated at 36 ± 1 °C for 48 h to facilitate colony counting. For species identification, PCR amplification was performed using specific primers targeting conserved regions of the 16S rRNA gene, which allowed for the accurate identification of microbial species present in the samples.

Aseptic techniques were followed during sampling, and the water flow from the tap was maximized for 3–5 min before sampling. Trained personnel from the professional laboratory department conducted the testing procedures. Bacterial counts in water collected before aerator and on the aerator were required to be ≤100 CFU/mL, and in final rinsing water, ≤10 CFU/100 mL, to meet acceptable medical water standards (*International Organization for Standardization, 2024*).

## Statistical analysis

Statistical analyses were performed using R 4.2.0 (*R Core Team, 2025*). The Fisher exact probability method was utilized to test the hypothesis concerning the success ratio (passedQC) and failure ratio (total–passedQC). In this approach, the number of successful events was denoted as passedQC, and the total number of observations or experiments was represented as total. A two-tailed test was conducted using the null hypothesis of equal
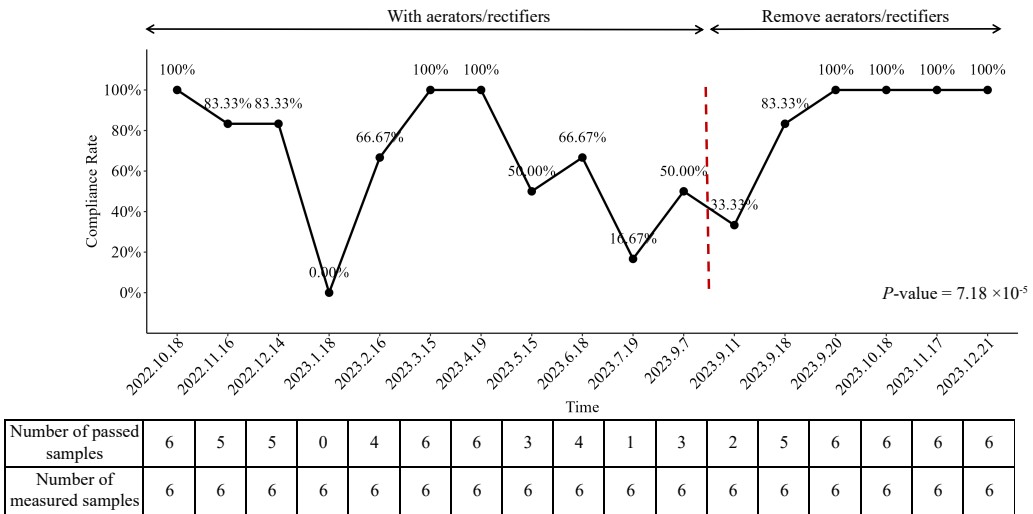

**Figure 2  Compliance rates of endoscopy final rinsing water from October 2022 to December 2023.**

proportions, comparing the proportion of positive samples in the initial test (October 2022) (6/6 samples) with the proportions in subsequent samples. A significance level of $P < 0.05$ was considered statistically significant.

# RESULTS

## Decrease in the compliance rate of endoscopy final rinsing water after hospital reconstruction

We analyzed the compliance rate of endoscopy final rinsing water from October 2022 to December 2023 and identified a statistically significant variation across the study period ($P$-value $= 7.18 \times 10^{-5}$). As illustrated in Fig. 2, the compliance rate was 100% (6/6 samples) in October 2022, prior to hospital reconstruction. However, following the completion of construction works, a marked decline in water quality was observed, with the compliance rate falling to 0% (0/6 samples) in January 2023.

In response to this deterioration, enhanced waterline disinfection was promptly implemented. This intervention was associated with a temporary improvement in compliance, with rates returning to 100% in March and April 2023. Despite this initial success, the compliance rate began to decline again thereafter, reaching 16.67% (1/6 samples) in July 2023, demonstrating a statistically significant decrease compared to infection control standards (Fig. 2).

## Faucet aerator impact on endoscopy final rinsing water compliance fluctuations

The above-mentioned findings indicated issues with the water quality. A comprehensive on-site investigation was conducted at the hospital, which confirmed the alignment of pipeline direction with the architectural drawings and the inspection methods and adherence to the required inspection methods and environmental standards. However,

**Table 1  The number of bacteria detected in the endoscopy rinse water and the type of colony detected.**

| | Water collected before aerator[a] (unit: CFU/mL) | The surface of aerator[a] (unit: CFU/mL) | Final rinsing water[b] (unit: CFU/100 mL) |
|---|---|---|---|
| Location 1 | – | 1,200 (*Cupriavidus pauculus*) | 15 (*Cupriavidus pauculus*) |
| Location 2 | – | 1,000 (*Cupriavidus pauculus*) | 45 (*Cupriavidus pauculus*) |
| Location 3 | – | 3,500 (*Cupriavidus pauculus*) | 10 (*Cupriavidus pauculus*) |
| Location 4 | – | – | – |
| Location 5 | – | 40,000 (*Cupriavidus pauculus*) | 160 (*Stenotrophomonas maltophilia* & *Cupriavidus pauculus*) |
| Location 6 | – | – | – |

**Notes.**

[–] The colony was not detected and passed the quality control assessment.

[a] Detection object was the aerators of the water pipe, and pouring method was used.

[b] Detection object was final rinsing water, and filter membrane method was used.

a notable accumulation of grime on the surfaces of aerator was observed, suggesting a potential association with the water quality issues.

In response, a comprehensive investigation was conducted on microbial contamination levels (CFU/mL) and species distribution in the water supply system, with sampling at critical points: final rinsing water, water collected before aerator, and the surface of aerator. A total of 18 water samples were obtained on September 7th, 2023, with eight samples (44.44%) showing the presence of colonies (Table 1). We found that water collected before aerator showed no colony presence, indicating a compliance rate of 100% (6/6 samples) (Table 1). However, among the eight non-compliant water samples, four were obtained from the surface of aerator and four were obtained from the final rinsing water (Table 1). Notably, the non-compliant water sample from the surface of aerator corresponded to the non-compliant final rinsing water from the same location (Table 1). These findings consistently supported the hypothesis that contaminate aerator might be the underlying cause of the fluctuating compliance rate of final rinsing water. The presence of colonies predominantly identified as cupriavidus pauculus in the water samples obtained from the surface of aerator and the corresponding non-compliant final rinsing water samples further strengthens this conclusion (Table 1). These results highlight the importance of regular cleaning and maintenance of aerator to ensure the quality and safety of endoscopy final rinsing water.

## Assessment of the effect of the implemented preventive measures

Following the removal of aerator from the faucets of the water supply system on September 10th, 2023, primary disinfection was implemented. On September 11th, 2023, the average colony number of aerators filtered water was measured at 38.33 CFU/100 mL (Fig. 3). Subsequently, after the implementation of secondary disinfection on September 15th, 2023, the test results on September 18th, 2023, demonstrated a statistically significant decrease in the average colony number of final rinsing water to 13.33 CFU/100 mL (Fig. 3). The compliance rate was correspondingly increased to 83.33% (5/6 samples), which was close to the standard of infection controls (100%, $P$-value = 1.00) (Fig. 2). Furthermore,

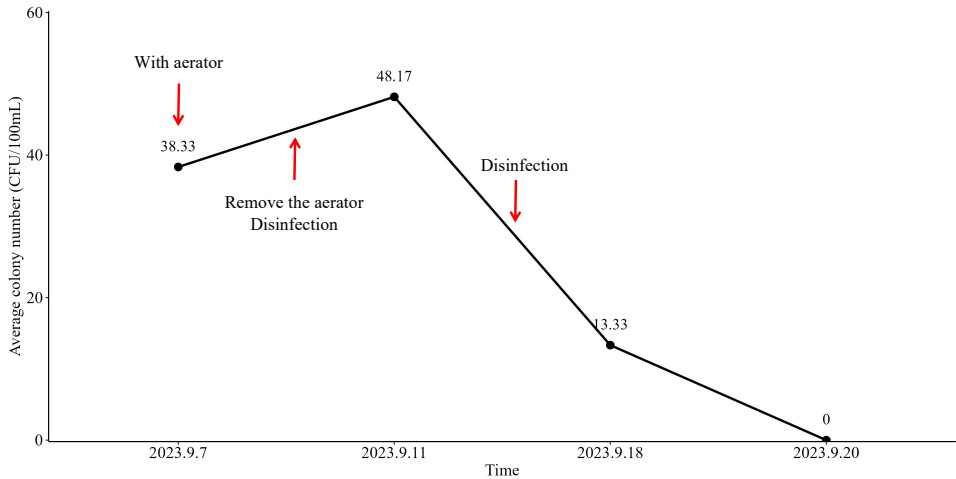

**Figure 3** **The number of bacteria detected in the final rinsing water of the manual tank in the center of the endoscope by the filter membrane method (unit: CFU/100 mL).**

on September 20th, 2023, the final rinsing water did not detect the presence of any colonies, resulting in an average colony number of 0 (Fig. 3). The compliance rate reached 100% (6/6 samples) (Fig. 2). These findings indicated that the implemented preventive measures, including the removal of aerator and the implementation of multiple disinfection steps, had a significant positive effect on the improvement of the compliance rate of endoscopy final rinsing water.

## DISCUSSION

Maintaining consistent compliance with microbial standards for final rinse water remains a critical but often overlooked challenge in endoscope reprocessing. Although international guidelines emphasize the importance of high-quality water to prevent healthcare-associated infections, real-world practice frequently reveals variability in compliance rates across clinical settings (*Ministry of Health of the People's Republic of China, 2022*; *International Organization for Standardization, 2024*; *National Health Family Planning Commission of the People's Republic of China, 2016*; *Speer et al., 2023*). Our study specifically investigated such fluctuations in a tertiary hospital and identified faucet aerators as a previously underappreciated source of microbial contamination. These findings align with a growing body of evidence suggesting that biofilm-prone components in hospital water systems can compromise disinfection outcomes, even when standard protocols are followed (*Takajo et al., 2020*).

Our investigation revealed a clear temporal and spatial association between aerator installation and fluctuations in microbial compliance, consistent with previous reports highlighting the role of aerators as biofilm reservoirs in hospital water systems (*Takajo et al., 2020*). Specifically, microbial colonization was detected on the surface of aerators, and bacterial species such as *Cupriavidus pauculus* and *Stenotrophomonas maltophilia* were isolated from final rinse water samples. These findings align with studies showing that

*Stenotrophomonas maltophilia* is frequently isolated from moist environments in healthcare settings and has been implicated in outbreaks of nosocomial infections, particularly among immunocompromised patients (*Khardori et al., 1990*; *Guyot, Turton & Garner, 2013*; *Alfieri et al., 1999*). Its presence in rinse water raises concerns about the potential for pathogen transmission during endoscopic procedures, emphasizing the need for continuous microbial surveillance and targeted risk mitigation strategies.

Biofilm formation likely played a central role in the persistence of microbial contamination despite regular disinfection. Biofilms are known to form rapidly on surfaces exposed to water and can significantly reduce the efficacy of disinfectants by limiting their penetration and contact time (*Otter et al., 2015*; *Bridier et al., 2011*; *Akinbobola et al., 2017*). In our study, weekly disinfection using chlorinated effervescent tablets at 500 mg/L failed to consistently achieve microbial compliance, suggesting that standard disinfection protocols may not be sufficient when biofilm colonization is present. These results are consistent with previous studies demonstrating that biofilms can serve as persistent sources of microbial contamination in endoscope reprocessing units and hospital water systems (*Takajo et al., 2020*).

The removal of aerators resulted in a marked and sustained improvement in water quality, with all samples collected after the intervention meeting regulatory standards ($\leq$10 CFU/100 mL). This simple intervention underscores the importance of identifying and eliminating known sources of microbial colonization in endoscopy units. These findings indicate that in settings where biofilm formation is a concern, eliminating potential colonization sites—such as aerators—may be a simple yet effective strategy to enhance infection control during endoscopic procedures.

This study acknowledges several limitations that should be considered when interpreting the results. Firstly, the retrospective design of the study introduces the potential for biases and limits the ability to establish a definitive causal relationship between the implemented interventions and the fluctuations in medical water compliance. Nevertheless, the consistent improvement in compliance rates following the removal of aerators provides strong observational evidence supporting their role as a key source of microbial contamination. Secondly, the study was conducted in a single tertiary hospital, which may limit the generalizability of the findings to other healthcare settings with different infrastructures or water management practices. Lastly, while PCR-based assays were employed to identify specific bacterial species in water samples, the scope of microbial characterization was limited in terms of strain-level resolution and functional profiling. The integration of more comprehensive molecular tools, such as whole-genome sequencing or metagenomic analysis, could further enhance our understanding of microbial ecology in rinse water systems and support the development of more precise and effective infection control strategies.

## CONCLUSIONS

This study underscore the impact of aerator on the compliance rate of medical water. Further research and guidelines are warranted to ensure appropriate usage of aerator, ultimately promoting patient safety and maintaining water quality standards.

### Funding

The authors received no funding for this work.

### Competing Interests

The authors declare there are no competing interests.

### Author Contributions

- Yuhua Yuan conceived and designed the experiments, performed the experiments, authored or reviewed drafts of the article, and approved the final draft.
- Lihong Ye performed the experiments, analyzed the data, prepared figures and/or tables, and approved the final draft.
- Tianyi Lu performed the experiments, authored or reviewed drafts of the article, and approved the final draft.
- Baihuan Feng analyzed the data, prepared figures and/or tables, and approved the final draft.
- Jin Zhao conceived and designed the experiments, authored or reviewed drafts of the article, and approved the final draft.

### Data Availability

The raw data is available in the Supplemental File.

### Supplemental Information

Supplemental information for this article can be found online at http://dx.doi.org/10.7717/peerj.20134#supplemental-information.

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
