# Peer review of "Factors influencing compliance with endoscopy final rinsing water standards: a study in a tertiary hospital setting"

_PeerJ, doi:10.7717/peerj.20134_

## Round 0.1 · original submission · Major Revisions

**Language Note:** The review process has identified that the English language must be improved. PeerJ can provide language editing services - please contact us at [email protected] for pricing (be sure to provide your manuscript number and title). Alternatively, you should make your own arrangements to improve the language quality and provide details in your response letter. – PeerJ Staff

Reviewer 1 ·

Basic reporting

Basic lab tests (agar plates and filtering) were performed that identified the presence and CFUs of two microbial species in rinse water over 15 months at a single hospital. Some CFU data are also presented. This is interesting data addressing an important topic (endoscope cleaning). Including at least a year of data is helpful. I do have a few concerns about harvesting, stat analyses, and how the results are reported - see comments under Experimental Design and Validity of Findings.

Minor Comments:
5. Line 38, I don’t think it is necessary to indicate the stat software here in the abstract.
6. Lines 67-8, is it helpful to also mention ISO 15883?
7. Lines 88-89 – it is not clear why all the data were not included? If you only look at the first test, what are you really studying?
8. Why were CFUs not collected directly? That would have substantially increased the usefulness of these data.
9. Line 124, what 2 proportions? From Figure 2 it appears that you are comparing to Jan 2022 proportion = 6/6. Please state that here.
10. Lines 129-133 – Are you saying that the large decrease in Jan 2023 wasn’t noticed until July 2023? Please explain why so long
11. Line 135 – what is the standard for disinfection?
12. Line 145, what do you mean by “colony conditions”?
13. Line 121. When referencing R, please cite and add to bibliography
R Core Team (2025). R: A Language and Environment for Statistical Computing. R Foundation for Statistical Computing, Vienna, Austria. <https://www.R-project.org/>.citation()
14. Why isn’t the detection of stenotrophomonas maltophilia mentioned in the abstract?
15. In Figure 2, earlier the x-axis only specifies year.month, later the x-axis specifies year.month.day. A consistent format should be used.

Experimental design

1. Lines 109-110, what validation was performed or what literature exists to support that gentle shaking in elution fluid does a good job of harvesting microbes from the aerator? Many times, sonication is used to enhance harvesting.
2. Table 1 says you detected Cupriavidus pauculais and stenotrophomonas maltophilia. If you did not use sequencing or PCR (line 207), how did you identify these species?

Validity of the findings

3. Reporting CFU/mL from the “surface of the aerator” in Table 1 doesn’t make any sense unless you give the volume of the elution fluid that the aerator was put into and then gently shaked. I suggest you transform the units to CFU/aerator.
4. There is an inflated Type 1 error. The p-values reported in Figure 2 appear to have been generated by 16 Fishers exact tests on 16 2x2 tables, all with an individual alpha=0.05, which increases the family-wise alpha = 1 – 0.95^16 = 0.56. That is, even if there are no differences between Jan 2022 and one of the 16 dates later on, there is a 56% chance that you will incorrectly find at least one of the 16 dates statistically significantly different from Jan 2022 (ie P-value < 0.05). One way to alleviate this issue is to perform Fishers test on a 2x17 table using the data in the table in Figure 2 (see ?fisher.test in R)

Reviewer 2 ·

Basic reporting

The manuscript is generally readable but contains awkward phrasing and grammatical errors (e.g. “cupriavidus pauculais” should be Cupriavidus pauculus; Line 41).
While the introduction rightly emphasizes the critical importance of endoscope rinse-water quality, its supporting references are problematic. Reference 2 does not document an outbreak, and references 3 and 4 discuss MDRO-related mortality without any direct link to endoscope transmission. The key outbreak studies in References 5–9 are largely outdated (some over 30 years old). Moreover, the manuscript’s reprocessing description is superficial and fails to acknowledge that current best practice and guideline recommendations favor automated endoscope reprocessors rather than the manual disinfection method apparently used at this center. The authors fail to reference any current international guidelines on this topic (Line 67). The manuscript also omits any discussion of biofilm formation, a key driver of water-related contamination.

Experimental design

The difference in acceptable bacterial counts between water sampled before the aerator and after it seems odd. To my knowledge, an aerator does not filter bacteria, and guidelines require final rinse water to be completely sterile. Additionally, the authors do not specify how frequently the aerators are replaced as part of routine maintenance.

Validity of the findings

The authors report percentages without providing the underlying counts, which undermines the clarity and interpretability of their findings. Line 142 describes a notable accumulation of grime on the aerator surfaces, suggesting non-compliance with cleaning protocols that could explain the final rinse-water contamination. Was this addressed in the cleaning procedures, and did the center intensify maintenance surveillance following this observation? The Discussion largely restates the Introduction and re-reports results instead of contextualizing them within the existing literature.

Additional comments

The scientific novelty is limited: the intervention was applied only once and not monitored over time. It would be more informative to determine the optimal aerator-replacement frequency to maintain compliance, or to assess whether an enhanced cleaning protocol could have prevented contamination.

---

## Round 0.2 · Minor Revisions

**Language Note:** When you prepare your next revision, please either (i) have a colleague who is proficient in English and familiar with the subject matter review your manuscript, or (ii) contact a professional editing service to review your manuscript. PeerJ can provide language editing services - you can contact us at [email protected] for pricing (be sure to provide your manuscript number and title). – PeerJ Staff

Reviewer 1 ·

Basic reporting

The authors have satisfactorily addressed my original comments regarding basic reporting.

Experimental design

The authors have satisfactorily addressed my original comments regarding experimental design.

Validity of the findings

The authors have satisfactorily addressed my original comments regarding the validity of findings, except for 1, #3: Reporting CFU/mL from the “surface of the aerator” in Table 1 doesn’t make any sense unless you give the volume of the elution fluid that the aerator was put into and then gently shaken. I suggest you transform the units to CFU/aerator.

The authors responded: We have now clarified in the Methods section (Page 5, Lines 122-123) that each aerator was immersed in a defined volume of elution fluid and gently shaken to recover surface microorganisms, which supports our reporting of results in CFU/mL.

From the authors' response and paper, it is still not clear what the “defined volume of elution fluid” is. The “defined volume of elution fluid” needs to be made clear in the paper.

Additional comments

The authors have satisfactorily addressed my original "additional comments" except 1, #7: Lines 88-89 – it is not clear why all the data were not included? If you only look at the first test, what are you really studying?

The authors responded: We selected only the initial test data for analysis because it reflects the baseline microbial status of the water before any corrective actions were taken. In routine monitoring, repeated sampling and interventions are performed until compliance is achieved, and including all data points would confound the assessment of the original contamination source. Therefore, using the first test results provides a clearer and more accurate representation of the actual microbial burden prior to intervention.

However, it does not appear that any of this explanation was incorporated into the revised manuscript. My strong opinion is that the authors add these clarifying statements to the manuscript.

---

## Round 0.3 · accepted · Accept

Thank you for revising your manuscript to address the reviewers' concerns. Reviewer 1 now recommends acceptance, and I am also satisfied with your response to the earlier comments of Reviewer 2. The manuscript is now ready for publication.

Reviewer 1 ·

Basic reporting

The authors have satisfactorily addressed all comments from my two reviews of their paper.

Experimental design

-

Validity of the findings

-